# Efficacy and Safety of Angiotensin Receptor Blockers in a Pre-Clinical Model of Arrhythmogenic Cardiomyopathy

**DOI:** 10.3390/ijms232213909

**Published:** 2022-11-11

**Authors:** Maicon Landim-Vieira, Aida Rahimi Kahmini, Morgan Engel, Elisa Nicole Cannon, Nuria Amat-Alarcon, Daniel P. Judge, José Renato Pinto, Stephen P. Chelko

**Affiliations:** 1Department of Biomedical Sciences, College of Medicine, Florida State University, Tallahassee, FL 32306, USA; 2Department of Nutrition and Integrative Physiology, Florida State University, Tallahassee, FL 32306, USA; 3Department of Medicine, School of Medicine, Johns Hopkins University, Baltimore, MD 21215, USA; 4Department of Medicine, Medical University of South Carolina, Charleston, SC 29425, USA

**Keywords:** arrhythmogenic cardiomyopathy, ARBs, PPARs, fibrosis

## Abstract

Arrhythmogenic Cardiomyopathy (ACM) is a familial heart disease, characterized by contractile dysfunction, ventricular arrhythmias (VAs), and the risk of sudden cardiac death. Currently, implantable cardioverter defibrillators and antiarrhythmics are the mainstays in ACM therapeutics. Angiotensin receptor blockers (ARBs) have been highlighted in the treatment of heart diseases, including ACM. Yet, recent research has additionally implicated ARBs in the genesis of VAs and myocardial lipolysis via the peroxisome proliferator-activated receptor gamma (PPARγ) pathway. The latter is of particular interest, as fibrofatty infiltration is a pathological hallmark in ACM. Here, we tested two ARBs, Valsartan and Telmisartan, and the PPAR agonist, Rosiglitazone, in an animal model of ACM, homozygous *Desmoglein-2* mutant mice (*Dsg2*^mut/mut^). Cardiac function, premature ventricular contractions (PVCs), fibrofatty scars, PPARα/γ protein levels, and PPAR-mediated mRNA transcripts were assessed. Of note, not a single mouse treated with Rosiglitazone made it to the study endpoint (i.e., 100% mortality: n = 5/5). Telmisartan-treated *Dsg2*^mut/mut^ mice displayed the preservation of contractile function (percent ejection fraction [%EF]; 74.8 ± 6.8%EF) compared to Vehicle- (42.5 ± 5.6%EF) and Valsartan-treated (63.1 ± 4.4%EF) mice. However, Telmisartan-treated *Dsg2*^mut/mut^ mice showed increased cardiac wall motion abnormalities, augmented %PVCs, electrocardiographic repolarization/depolarization abnormalities, larger fibrotic lesions, and increased expression of PPARy-regulated gene transcripts compared to their *Dsg2*^mut/mut^ counterparts. Alternatively, Valsartan-treated *Dsg2*^mut/mut^ mice harbored fewer myocardial scars, reduced %PVC, and increased Wnt-mediated transcripts. Considering our findings, caution should be taken by physicians when prescribing medications that may increase PPARy signaling in patients with ACM.

## 1. Introduction

Arrhythmogenic cardiomyopathy (ACM) is an inherited heart disease associated with progressive muscle deterioration, lethal arrhythmias, exercise-induced disease penetrance, inflammation, and fibrofatty replacement of the myocardium [1]. Known as a “disease of the cardiac desmosome,” the majority of clinical reports are linked to inherited pathogenic variants (primarily, autosomal dominant) in desmosomal genes, such as *PKP2* (*Plakophilin-2*) and *DSG2* (*Desmoglein-2*) [1]. Despite experimental and clinical evidence on the efficacy of current pharmacological treatment options, such as antiarrhythmics and angiotensin receptor blockers (ARBs), little is known regarding ARBs’ role in the origin of ventricular arrhythmias and increased myocardial lipolysis via the peroxisome proliferator-activated receptor (PPAR) pathway. Kim, C. et. al. demonstrated in induced pluripotent stem-cell-derived cardiomyocytes (iPSC-CMs) from a patient harboring a pathogenic *PKP2* variant that introducing exogenous PPARγ caused a shift in metabolic respiration, increased lipolysis and apoptosis, decreased β-catenin activity, and abnormal calcium handling [2]. In the present study, we used an animal model of ACM-homozygous *Desmoglein-2* mutant mice (*Dsg2*^mut/mut^) [3,4,5]. This is a robust animal model of ACM that recapitulates the disease time course seen in patients with ACM. Specifically, *Dsg2*^mut/mut^ mice are asymptomatic at birth and show subclinical phenotypes (e.g., ECG depolarization/repolarization abnormalities) until 8 weeks of age [3,4,5] whereby, at 8 weeks of age, echocardiographic analyses demonstrate mild dysfunction [3,4,5]. From 8 to 16 weeks of age, large-scale myocyte cell death and both innate and extrinsic inflammation lead to pathological progression [3,4,5], whereby, at 16 weeks of age, *Dsg2*^mut/mut^ mice demonstrate numerous PVCs, severe cardiac dysfunction, and biventricular fibrosis (Appendix A) [3,4,5].

Thus, *Dsg2*^mut/mut^ mice were utilized in this study to provide further insight into the efficacy and safety of two specific ARBs with discordant effects on PPAR signaling, Valsartan and Telmisartan, as a pharmacological therapeutic in ACM. Moreover, we treated mice with Rosiglitazone, a selective PPARγ agonist with no PPARα-binding affinity to test whether drugs with PPARγ agonism (i.e., Telmisartan or Rosiglitazone) worsen ACM phenotypes.

Recent studies demonstrated the deleterious effects of PPARγ signaling in the pathophysiology of ACM progression [2,6]. However, these prior studies lack in vivo testing of thiazolidinediones (i.e., PPARγ agonists) in animal models of ACM. Our study reveals the detrimental impact of PPARγ agonism in ACM mice, as a 100% mortality rate occurred in Rosiglitazone-treated *Dsg2*^mut/mut^ mice before the study endpoint. Yet, even at the study midpoint, Rosiglitazone-treated *Dsg2*^mut/mut^ mice had significant cardiac dysfunction. Caution is warranted when prescribing ARBs to ACM patients, such as those with comorbidities (e.g., diabetes mellitus type-2). Even so, thiazolidinediones have been known to cause fluid retention and thus may complicate treatment with ACM patients developing or with developed heart failure. Alternatively, while Valsartan-treated *Dsg2*^mut/mut^ mice did not delay the progression of cardiac dysfunction, it normalized QRS duration, reduced ventricular ectopy, and prevented further fibrotic progression. Interestingly, Valsartan-treated *Dsg2*^mut/mut^ mice showed increased nuclear PPARα levels and increased active β-catenin with elevated Wnt/β-catenin transcripts, the latter of which is down-regulated in ACM [7].

In summary, this article provides evidence that Valsartan is a safe and effective ARB to reduce ventricular ectopy and ECG repolarization and depolarization abnormalities, prevent fibrofatty scar formation, and increase Wnt-mediated gene transcription. More studies should be conducted regarding drug titration levels of Valsartan to acquire the optimal dose for the preservation of cardiac function without adverse outcomes. Although Telmisartan-treated *Dsg2*^mut/mut^ mice did show the preservation of cardiac function, caution should be implemented for providers in prescribing Telmisartan, especially considering our results indicate worsening of arrhythmic burden, a variety of cardiac wall motion abnormalities, and increased fibrotic remodeling in Telmisartan-treated *Dsg2*^mut/mut^ mice.

## 2. Results

To test the efficacy and safety of angiotensin receptor blockers (ARBs) in ameliorating or delaying ACM phenotypes, wild-type (WT) and *Dsg2*^mut/mut^ mice were provided either Valsartan (Val), Telmisartan (Tel), or Vehicle (Veh) via drinking water for 13 weeks starting at 3 weeks of age. At the study endpoint (i.e., 16 weeks of age, [16 W]), 100% of Val-treated *Dsg2*^mut/mut^ mice survived, whereas, at week 15, n = 1 Veh-treated *Dsg2*^mut/mut^ mouse died (overall cohort survival = 85.7%; Figure 1A). Interestingly, n = 1 Tel-treated *Dsg2*^mut/mut^ mouse died 3 weeks prior (overall cohort survival = 85.7%; Figure 1A) to the Veh-treated *Dsg2*^mut/mut^ mouse that died. No deaths occurred in Veh-treated WT mice (Figure 1A).

The damaging effects of PPARγ agonism in ACM have been previously demonstrated [2,6], yet to our knowledge, this has yet to be tested in vivo. Therefore, we treated *Dsg2*^mut/mut^ mice with Rosiglitazone (Ros), a thiazolidinedione that specifically acts as a PPARγ agonist. Of monumental importance, Ros-treated *Dsg2*^mut/mut^ mice exhibited a 100% mortality rate by 9 weeks of age (Figure 1A). Considering the rapid mortality rate in Ros-treated *Dsg2*^mut/mut^ mice, we collected Echos from the remaining n = 3 mice at 7W in the event that more deaths occurred in this cohort. We observed a stark decrease in cardiac function and aberrant cardiac remodeling in Ros-treated *Dsg2*^mut/mut^ mice vs. Veh-treated *Dsg2*^mut/mut^, Veh-treated WT, and Ros-treated WT cohorts at 7 weeks of age (Appendix A).

At the study midpoint (8W) and endpoint (16W) (Appendix A and Table 1, respectively), echocardiographic, ECG, and blood pressure plethysmography (BPP) data were collected. *Dsg2*^mut/mut^ mice treated with either Veh or Val presented with reduced cardiac function (%EF) at 16W compared to Veh-treated WT mice (Figure 1B; Table 1). That said, Val-treated *Dsg2*^mut/mut^ mice still showed improved cardiac function compared to Veh-treated *Dsg2*^mut/mut^ mice (Figure 1B, Table 1). Conversely, Tel-treated *Dsg2*^mut/mut^ mice showed the preservation of cardiac function compared to Veh- and Val-treated *Dsg2*^mut/mut^ counterparts (Figure 1B; Table 1). Despite this finding, Echo analyses in Tel-treated *Dsg2*^mut/mut^ mice demonstrated that this cohort presented with hypokinesia, dyskinesia, and aneurysms of the left ventricular anterior and posterior free walls (Figure 1C, D; Table 1). Next, signal-average ECGs (SAECGs) showed that Val- and Tel-treated *Dsg2*^mut/mut^ mice displayed shortening of the RR-Interval (RR-I) and increased heart rate (HR) compared to Veh-*Dsg2*^mut/mut^ mice (Figure 1E,F; Table 1). Although Veh-*Dsg2*^mut/mut^ mice showed a trend towards prolonged QRS-duration (QRSd), only Tel-treated *Dsg2*^mut/mut^ mice showed increased QRSd compared to WT mice (Figure 1F; Table 1). Additionally, Tel-treated *Dsg2*^mut/mut^ mice demonstrated augmented premature ventricular contractions (PVCs; Figure 1G) and an increased Q-amplitude (Figure 1E; Table 1); thus, Telmisartan appears to cause ventricular depolarization abnormalities in ACM mice. As a vasodilator, Telmisartan exhibited a nearly 30% reduction in systolic, diastolic, and mean blood pressure compared to Veh- and Val-treated *Dsg2*^mut/mut^ mice (Table 1). A similar reduction in blood pressure was observed in Tel-treated WT mice, yet Valsartan appeared to have no effect in WT mice.

Next, the cardiac histopathological analysis showed a significant increase in right ventricular fibrosis (RVF) in Veh-treated *Dsg2*^mut/mut^ mice compared to Veh-treated WT mice (Figure 2A,B). Although RVF was increased in Val-treated *Dsg2*^mut/mut^ mice compared to Veh-treated WT mice, this cohort displayed a reduction in RVF compared to Veh-treated *Dsg2*^mut/mut^ mice (Figure 2B). Meanwhile, Tel-treated *Dsg2*^mut/mut^ mice showed a null-to-minimal reduction in RVF (Figure 2A,B). To assess the influence of ARBs on PPARα/γ activation and PPAR-mediated transcription, immunoblots were performed in nuclear lysates and isolated mRNA, respectively. Veh-treated *Dsg2*^mut/mut^ mice displayed a trend towards reduced nuclear PPARα levels compared to Veh-treated WT mice (Figure 2C,E). Interestingly, nuclear PPARα levels were elevated in Val-treated *Dsg2*^mut/mut^ mice compared to Veh- and Tel-treated *Dsg2*^mut/mut^ mice (Figure 2C,E). Regardless of which ARB treatment, *Dsg2*^mut/mut^ mice displayed a reduction in nuclear PPARγ levels (Figure 2C,F) compared to WT mice, counter to our original hypothesis. Conversely, we examined the levels of β-catenin, considering PPARγ and Wnt act in opposition to one another. As previously reported in ACM, [2] *Dsg2*^mut/mut^ mice harbored a significant reduction in the level of active (non-phosphorylated) β-catenin (Figure 2D,G).

Lastly, ARB-induced gene transcription was analyzed for changes in PPARs, PPAR co-regulators, and metabolism and lipid transcripts (Figure 2H and Appendix A). The most interesting results uncovered were that both *Ppar_α_*, *Ppar_δ_,* and *PPARγ co-activator-1α* (*Pgc1α*) and -*1β* (*Pgc1β*) expression was elevated in Tel-treated *Dsg2*^mut/mut^ mice (Figure 2H). This was particularly interesting considering PPARγ nuclear protein levels were reduced in these same mice. Of note, *pyruvate dehydrogenase* (*Pdh*) and *Pdh kinase isoform-2* (*Pdk2*) expressions were up-regulated in Val-treated *Dsg2*^mut/mut^ mice compared to Veh-treated *Dsg2*^mut/mut^ mice (Appendix A). Intriguingly, the top transcription factor binding sites for *Pdh* and *Pdk2* gene promoters are *c-Myc* and *Nfat1-4*, whereby the former is a primordial Wnt/β-catenin transcript and the latter increases the transcription of the Frizzled—A Wnt co-receptor [8]. These findings are certainly intriguing given that Val-treated *Dsg2*^mut/mut^ mice showed elevated levels of β-catenin (*Ctnnb1*; Figure 2H). There was a stark decline in *Pdk4* expression, an endogenous inhibitor of *Pdh*, in Val-treated *Dsg2*^mut/mut^ mice compared to Veh-treated *Dsg2*^mut/mut^ mice (Appendix A). Although the expression of *adiponectin* (*Adipo*) was elevated in nearly all *Dsg2*-cohorts (Appendix A), only Val-treated *Dsg2*^mut/mut^ mice showed reduced levels of the adipokine, *fatty acid binding protein-4* (*Fabp4*; Appendix A). In conclusion, it appears Valsartan increases gene transcripts to catabolize fat into energy (i.e., ATP) through the TCA cycle, in addition to increasing Wnt/β-catenin transcription.

## 3. Discussion

Considering that fibrofatty replacement of the myocardium is a pathological hallmark of ACM, we were interested in the effects of thiazolidinediones (i.e., Rosiglitazone) and/or partial-thiazolidinediones (i.e., Telmisartan), on cardiac function, lipolytic gene expression, and fibrofatty infiltration. The ARB, Telmisartan, was of particular interest given recent work that demonstrated it harbors partial-PPARγ agonism [9]. This is crucial in ACM given that the suppression of the Wnt/β-catenin signaling pathway is implicated in ACM pathogenesis [7] and PPARγ and Wnt share an inverse relationship with one another. Specifically, PPARs work in complete opposition to Wnt, such that if PPARs are up-regulated then Wnt is down-regulated, and vice versa [10].

As an ARB, Telmisartan exceeded the blood pressure reduction compared to Valsartan at both 8 and 16 weeks of age. Additionally, Telmisartan preserved left ventricular function to nearly WT levels. However, Telmisartan-treated *Dsg2*^mut/mut^ mice showed increased wall motion abnormalities, myocardial fibrosis, PVCs, and depolarization abnormalities—all Task Force Criteria phenotypes for ACM. Alternatively, while Valsartan preserved cardiac function, it was nearly devoid of conduction abnormalities, arrhythmias, and reduced RVF, and myocardial samples showed increased PPARα nuclear levels, active β-catenin, and Wnt-mediated mRNA transcription. Prior work demonstrated PPARα activation is necessary to combat ROS-mediated cellular damage and inflammation via the suppression of NFκB [11]. Exercise-induced ROS generation [5] and NFκB-mediated myocardial inflammation [4] are two key pathological triggers we recently demonstrated in ACM.

In addition to either ARBs’ efficacy in preventing cardiac dysfunction and fibrosis, we would be remiss to not address the tantalizing outcomes of *Pdh* and *Pdk4* expression in Val-treated *Dsg2*^mut/mut^ mice. Pyruvate dehydrogenase catalyzes pyruvate and NAD+ into acetyl-CoA and NADH, the latter of which acts as a reducing agent for the mitochondrial Thioredoxin-2/Peroxiredoxin-3 pathway, a pathway we have shown is severely reduced in ACM myocardial samples [5]. Thus, it appears Valsartan not only inadvertently increases reducing agents (i.e., NADH) to combat ROS levels but also elevates acetyl-CoA levels and thus energy production through the TCA cycle. This is a cycle that has additionally been shown to be down-regulated in ACM myocytes [2]. Lastly, PDK4 inhibits PDH via phosphorylation, thereby contributing to the regulation of fatty-acid oxidation (FAO) via the reduction of glucose metabolism. Considering heart failure (HF) is common in ACM patients, it may seem counterintuitive that *Pdk4* expression was significantly down-regulated in Val-treated *Dsg2*^mut/mut^ mice. Heart failure is a disease state that utilizes glucose metabolism over FAO, which produces far less ATP. Yet, recent work demonstrated that altered cellular metabolism, from glucose metabolism to the TCA cycle, is mediated by the reduction in PDK4 levels [12,13]. That said, Sun Y et al. showed PDK4 binds to the apoptosis-inducing factor (AIF), a mitochondrial protein that we recently showed contributes to cell death in ACM [5].

Although HF shifts the heart to a less energy-efficient metabolic state, new advancements have indicated that increasing glucose utilization in cardiomyocytes is associated with increased cardiac function [12,13]. In conclusion, providers should consider moving alongside antiarrhythmics in ACM, such as the administration of Valsartan as this ARB demonstrates additional metabolic therapeutic effects to increase substrate utilization even in diseased states (i.e., HF) [12,13].

## 4. Materials and Methods

### 4.1. Study Approval

All experiments conformed to the Guide for the Care and Use of Laboratory Animals from the National Institute of Health (NIH publication no. 85–23, revised 1996). Animal study protocol was approved by the Johns Hopkins University Animal Care and Use Committee (Protocol Code: MO19M94; Date of Approval: 26 March 2019).

### 4.2. In Vivo Drug Treatment

Two mouse lines were utilized in this study: Wild-type (WT) controls and homozygous *Desmoglein-2* mutant mice (*Dsg2*^mut/mut^). All mouse lines are of a C57BL/6J background and were bred in-house via heterozygous x heterozygous pairing (i.e., *Dsg2*^mut/+^ x *Dsg2*^mut/+^) in order to ensure all mice were age- and litter-matched, with n≥5 mice/cohort/treatment [5]. As previously described, [3] mice with the targeted allele were mated with mice harboring ubiquitous expression of CMV-Cre to remove the fourth and fifth exons of *Dsg2* (*Dsg2*^mut/mut^), which results in the loss of exons 4 and 5 due to a frameshift mutation and premature termination of translation. [3] This led to germline deletion of exons four and five, thus no longer requiring additional Cre-recombinase.

Litters were genotyped at 3 weeks of age then separated into treatment cages and administered one of the following via drinking water: (a) Valsartan (30 mg/kg/day) [14] (Cat. No. SML0142, Sigma; Milwaukee, WI, USA); (b) Telmisartan (10 mg/kg/day) [15] (Cat. No. T8949, Sigma); (c) Rosiglitazone (25 mg/kg/day) [16] (Cat. No. R2408, Sigma); or (d) Vehicle (deionized [DI] H_2_0). All drinking water bottles contained 5% Splenda. Genotypes were known prior to the separation into treatment cohorts to ensure n ≥ 5 mice/cohort/treatment. ARBs and Rosiglitazone were obtained as ≥98% HPLC grade powder, dissolved in DMSO, where stock solutions were then added to DI H_2_0 containing 5% Splenda and filter-sterilized with Express PLUS Stericups (0.22 μm; Cat. No. SCGPCAPRE, Millipore Sigma; Milwaukee, WI, USA). Mice were given access to treatment or vehicle water bottles ad libitum, and water bottles were replaced every 48 h.

### 4.3. Cardiac Function and Histopathology

Disease functional characterization was performed at 8 and 16 weeks (8 W and 16 W, respectively) of age. Specifically, echocardiography, electrocardiography, and blood pressure plethysmography were obtained at 8 W and 16 W, and histopathology, PPAR protein levels, and qPCR expression were obtained at 16 W.

Echocardiography: A 2100 Vevo Visualsonic was used to assess cardiac function. An ultrahigh-frequency linear array microscan transducer (40 MHz; acquired at a sweep speed of 200 mm/s,) was utilized to obtain short-axis, m-mode and parasternal, long-axis, B-mode images. All images at the level of the papillary muscles were obtained and analyzed following the American Society of Echocardiography guidelines for animals, as previously described [5]. Each mouse had 3–5 measurements obtained for each echocardiographic parameter and then were averaged for statistical evaluation, as previously described [3]. Additionally, all mice were evaluated for the presence of hypokinesia, dyskinesia, and aneurysms at the study endpoint; these functional abnormalities and corresponding data are included in Table 1.

Electrocardiography (ECG)*:* Surface ECGs were performed in anesthetized mice (nose cone anesthesia using 2% Isoflurane in 100% vaporized oxygen [O_2_]). PowerLab (ADInstruments) instruments for Lead I recordings were utilized, as previously described [5]. ECG Analysis Add-on Software (LabChart Pro 8, MLS360/8, AD Instruments) was utilized to analyze signal-averaged ECGs (SAECGs; Appendix A and Table 1 for 8 W and 16 W, respectively). Additionally, 5 min recordings were used to obtain the percent ventricular extrasystoles (%PVCs).

Blood pressure plethysmography (BPP)*:* Non-invasive BPP measurements were obtained via the ADInstruments Tail-cuff BPP System. Specifically, non-anesthetized mice (a) were led into a rodent restrainer (Cat. No. MLA5016); (b) using the slide piston, each mouse was adjusted in order to maximize tail length for placement in the BPP tail cuff holder (Cat. No. MLA5030); (c) a blood pressure cuff (Cat. No. MLT125/m) was then placed around the base of the tail. All BPP measurements were conducted in a 37 °C designated quiet area (JHU Physiology Suite). Prior to BPP measurements, (d) mice were acclimated to the ADInstruments Tail-cuff BPP System for 15mins. Following acclimation, (e) a total of n = 6 BPP measurements were recorded. The first n = 3 measurements were deemed ‘acclimation cycles’ and discarded, whereas the last n = 3 measurements were averaged for systolic (SBP), diastolic (DBP), and mean BP (MBP) recordings.

Histopathology: Following endpoint functional analyses, mice were euthanized via cervical dislocation, their hearts were extracted and placed in 1X PBS, and they were cut via the long axis. One-half of the heart containing the atrium was processed for immunohistochemical analyses via Masson’s Trichrome, as previously described [5]. The remaining half was flash-frozen in liquid nitrogen for downstream analyses.

### 4.4. PPAR Pathway Analyses

Subcellular fractionation: Mouse ventricular myocardium was lysed in RIPA buffer containing 1:100 protease and phosphatase inhibitors (Sigma; Cat. No. P1860 and P0044, respectively). Following the removal of cytosolic and membrane-bound lysates, nuclear fractions were extracted as follows: 5000 g for 5 min at 4 °C (Subcellular Fractionation Kit for Tissues, ThermoFisher, Cat. No. 87790). Protein concentrations were then measured using a BCA protein assay kit (Pierce, Cat. No. 23225)

Western immunoblotting: Following subcellular fractionation, extracted nuclear fractions (10 µg) were boiled (95 °C) in Laemmli buffer for 10 min and separated via electrophoresis in NuPAGE^TM^ 4–12% Bis-Tris gels at 100 V. Protein-captured gels were then transferred to nitrocellulose membranes using the iBlot^TM^-2 system at 20 V for 7 min. Membranes were blocked using the iBind^TM^ flex FD solution (ThermoFisher; Cat. No. SLF2020) for 30 min at room temperature. Membranes were placed in the iBind^TM^ flex system and probed with the following primary antibodies: Mouse anti-active β-catenin (Sigma; Cat. No. 05-665; at 1:1000); rabbit anti- β-catenin (Sigma; Cat. No. 06-734; at 1:1000); mouse anti-Histone-3 (H3; Cell Signaling; Cat. No. 14269S; at 1:1000); rabbit polyclonal anti-PPARα (Abcam; Cat. No. 8934; at 1:500); and rabbit polyclonal anti-PPARγ (GeneTex; Cat. No. 50828; at 1:500). To avoid signal overlap, immunoblots for PPARα and PPARγ were performed separately. Secondary antibodies were additionally added as follows: (a) Either added to the iBind^TM^ flex system: Goat anti-rabbit IRDye 680RD (LI-COR^TM^; Cat. No. 926-6807 at 1:1000); or blots were incubated at room temperature for 1 h with donkey anti-mouse IRDye 800CW (LI-COR^TM^; Cat. No. 926-32212; at 1:1000 dilution). Membranes were then washed three times (5 min each) with 1X TBS with 0.05% Tween^TM^-20 and imaged using the Odyssey^TM^ CLx imaging system (LI-COR^TM^). Protein signal intensities (arbitrary units; [a.u.]) were quantified using the Odyssey Imaging system (Image Studio 4.0 software). Protein signals corresponding to nuclear PPARα/γ were normalized to H3 signal intensities, then cohorts were normalized to Veh-treated WT signals.

### 4.5. qPCR Expression

RNA (1 µg/reaction) was isolated using the PicoPure RNA Isolation Kit (ThermoFisher, Cat. No. KIT0204) and cDNA (0.5 µg/reaction) generated using the High-capacity cDNA Reverse Transcription Kit (Fisher Scientific, Cat. No. 43-688-14; Foster City, CA, USA). The SYBR Green Select Master Mix (ThermoFisher, Cat. No. 4472919; Gainesville, FL, USA) was utilized for qPCR reactions in the CFX384 Biorad Real-Time PCR System (BioRad; Cat. No. 1855484). All reactions were normalized to *Gapdh* and then normalized to Veh-treated WT data. Forward and reverse qPCR primers and protocol information can be found in Appendix A.

### 4.6. Statistical Analyses

Data are presented as mean ± SEM, with a *p* < 0.05 deemed significant using Student’s t-test and One-way or Two-way ANOVA with Tukey’s post-hoc analysis. A detailed description of statistical analyses can be found in each figure legend and/or table. The number of animals used (n = 5–6) in this study was determined based on our prior publications [3,4,5] that demonstrated n ≥ 4 mice was sufficient to show substantial cardiovascular and pathological differences between cohorts (i.e., WT vs. *Dsg2*^mut/mut^). Our prior work demonstrated 80–99% Power Analyses and received a statistical probability of *p* < 0.05 by utilizing an n ≥ 4 mice/genotype/cohort/parameter.

## Figures and Tables

**Figure 1 ijms-23-13909-f001:**
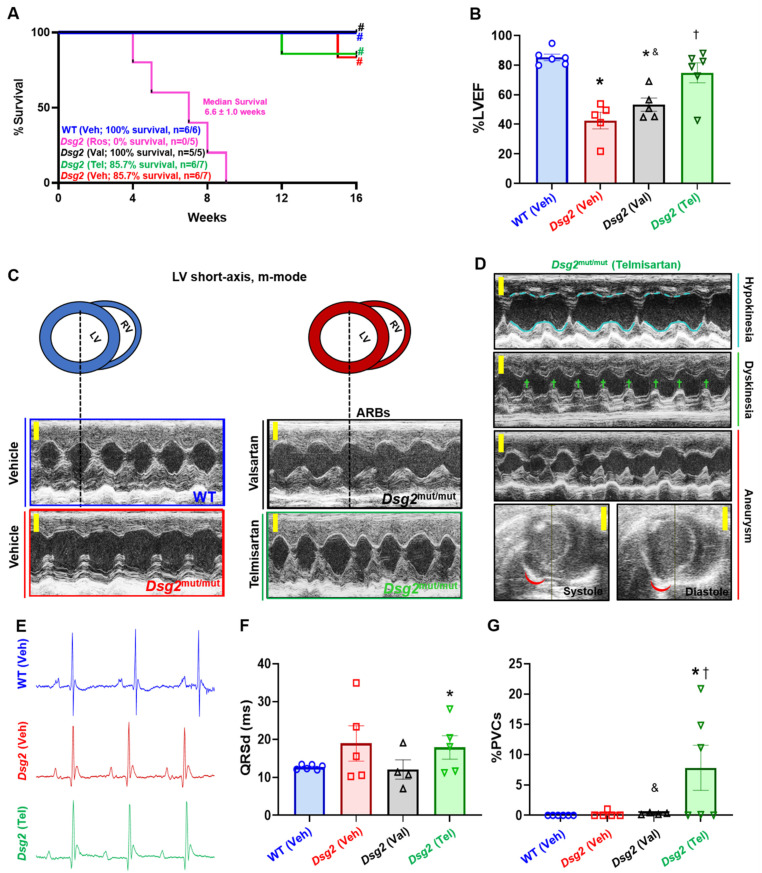
Survival and cardiac function in *Dsg2*^mut/mut^ mice treated with ARBs. (**A**) Percent survival. (**B**) Percent left ventricular ejection fraction (%LVEF). (**C**) Representative short-axis, m-mode echocardiograms from Vehicle- and ARB-treated cohorts. (**D**) Hypokinesis of the LV anterior wall (dashed blue lines) with normokinesis of the LV posterior wall (solid blue lines). Dyskinesia of the LV posterior wall (green crosses). LV posterior free wall aneurysm proximal to the interventricular septum (red arc). For (**C**) and (**D**), yellow scale bars, 2 mm. (**E**) Representative electrocardiography tracings taken in Lead I. Note, increased Q-amplitude, reduced S-amplitude, and shortening of the RR-Interval (RR-I) in Vehicle- and Telmisartan-treated *Dsg2*^mut/mut^ mice. (**F**) QRS duration (QRSd). (**G**) Percent premature ventricular contractions (%PVCs). Data presented as mean ± SEM. * *p* < 0.05 for any cohort vs. Vehicle-treated WT mice; ^†^
*p* < 0.05 for any drug-treated *Dsg2*^mut/mut^ cohort vs. Vehicle-treated *Dsg2*^mut/mut^ mice; ^&^
*p* < 0.05 for Valsartan-treated *Dsg2*^mut/mut^ mice vs. Telmisartan-treated *Dsg2*^mut/mut^ mice, using One-way ANOVA with Tukey’s post-hoc analysis. ^#^
*p* < 0.05 any cohort vs. Rosiglitazone-treated *Dsg2*^mut/mut^ mice using Mantel-Cox and Wilcoxon survival analysis. Blue, WT vehicle-treated mice; pink, *Dsg2*^mut/mut^ Rosiglitazone-treated mice; black, *Dsg2*^mut/mut^ Valsartan-treated mice; green, *Dsg2*^mut/mut^ Telmisartan-treated mice; and red, *Dsg2*^mut/mut^ vehicle-treated mice.

**Figure 2 ijms-23-13909-f002:**
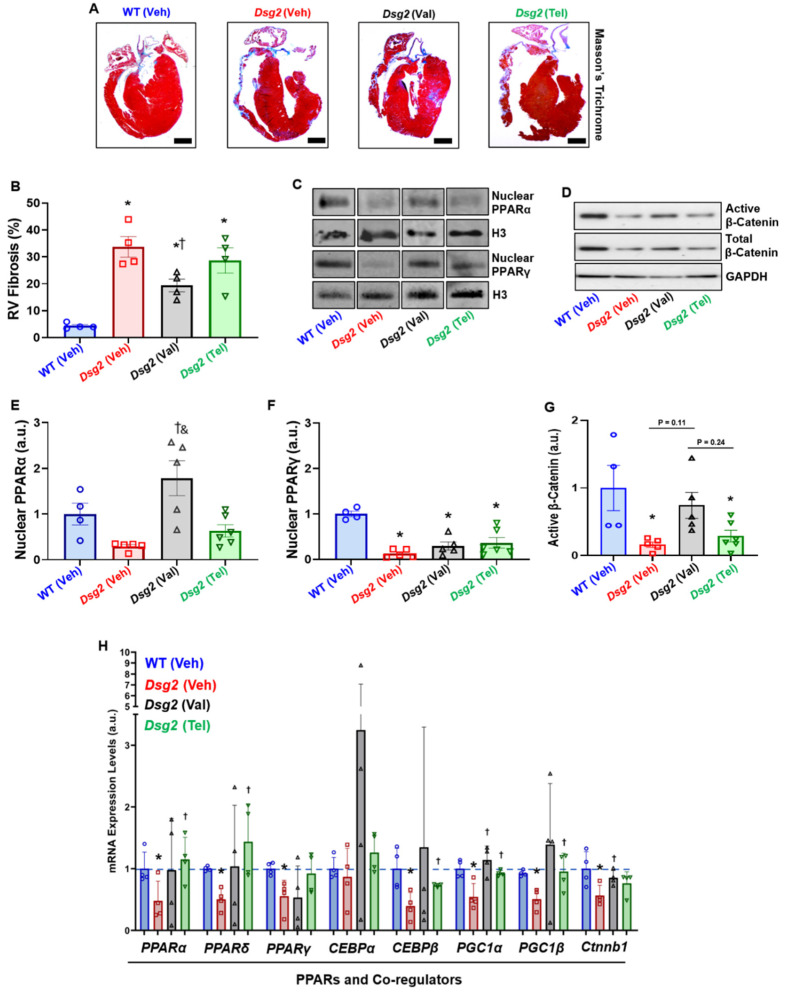
Cardiac histopathological analysis and nuclear PPARα and PPARγ levels in Vehicle-treated WT and ARB-treated *Dsg2*^mut/mut^ mice. (**A**) Representative cardiac tissue slices stained with Masson’s Trichrome for fibrotic lesions from Vehicle-treated WT mice and *Dsg2*^mut/mut^ mice treated with ARBs. Black scale bar, 1 mm. (**B**) Percent fibrosis of the right ventricle (RV) free wall. (**C**,**D**) Representative immunoblots probed for PPARγ, PPARα, active β-Catenin, total β-Catenin, and Histone-3 (H3). Immunoblot quantification of nuclear (**E**) PPARα, (**F**) PPARγ, and (**G**) active β-Catenin in Vehicle-treated WT mice and *Dsg2*^mut/mut^ mice treated with ARBs. (**H**) qPCR analysis of genes associated with PPARs and co-regulators. Data presented as mean ± SEM. * *p* < 0.05 for any cohort vs. Vehicle-treated WT mice; ^†^
*p* < 0.05 for any drug-treated *Dsg2*^mut/mut^ cohort vs. Vehicle-treated *Dsg2*^mut/mut^ mice; ^&^
*p* < 0.05 for Valsartan-treated *Dsg2*^mut/mut^ mice vs. Telmisartan-treated *Dsg2*^mut/mut^ mice One-way ANOVA with Tukey’s post-hoc analysis. Blue, WT vehicle-treated mice; pink, *Dsg2*^mut/mut^ Rosiglitazone-treated mice; black, *Dsg2*^mut/mut^ Valsartan-treated mice; green, *Dsg2*^mut/mut^ Telmisartan-treated mice; and red, *Dsg2*^mut/mut^ vehicle-treated mice.

**Table 1 ijms-23-13909-t001:** Echocardiographic, electrocardiographic, blood pressure plethysmography, and morphometric indices from Vehicle- and Drug-treated WT and *Dsg2*^mut/mut^ mice at 16 weeks of age. IVSd, interventricular septal end-diastolic volume; IVSs, interventricular septal end-systolic volume; LVIDd, left ventricular internal diameter end-diastolic volume; LVIDs, left ventricular internal diameter end-systolic volume; LVPWd, left ventricular posterior wall end diastole; LVPWs, left ventricular posterior wall end systole; FS, fractional shortening; EF, ejection fraction; HR, heart rate; RR-I, R-R interval; PR-I, P-R interval; Pd, P-wave duration; QRSd, QRS duration; Q-Amp, Q-wave amplitude; S-Amp, S-wave amplitude; PVC, premature ventricular contraction; BPP, blood pressure plethysmography; SBP, systolic blood pressure; DBP, diastolic blood pressure; MBP, mean blood pressure; RWT, relative wall thickness; LVM, left ventricular mass; LVW/BW, left ventricular weight to body weight; LW/BW, liver weight to body weight; HW/BW, heart weight to body weight. Data presented as mean ± SEM, * *p* < 0.05 for any cohort vs. Vehicle-treated WT mice; ^†^
*p* < 0.05 any drug-treated *Dsg2*^mut/mut^ cohort vs. Vehicle-treated *Dsg2*^mut/mut^ mice; ^&^
*p* < 0.05 for Valsartan-treated *Dsg2*^mut/mut^ mice vs. Telmisartan-treated *Dsg2*^mut/mut^ mice using One-way ANOVA with Tukey’s post-hoc analysis.

Parameters	Vehicle		Valsartan (30 mg/kg/day)		Telmisartan (10 mg/kg/day)	
	WT	*Dsg2* ^mut/mut^	WT	*Dsg2* ^mut/mut^	WT	*Dsg2* ^mut/mut^
**Echo**						
*n*	6	5	5	5	5	6
IVSd (mm)	0.96 ± 0.03	0.91 ± 0.09	0.79 ± 0.05 *	0.84 ± 0.05 *	0.74 ± 0.06 *	0.73 ± 0.08 *
IVSs (mm)	1.52 ± 0.06	1.39 ± 0.07	1.38 ± 0.08	1.30 ± 0.04 *	1.36 ± 0.06 *	1.24 ± 0.05 *^†^
LVIDd (mm)	2.46 ± 0.14	2.96 ± 0.17 *	2.55 ± 0.19	2.90 ± 0.14 *	2.65 ± 0.31	2.61 ± 0.16
LVIDs (mm)	0.95 ± 0.13	2.23 ± 0.18 *	1.26 ± 0.17	1.98 ± 0.15 *^&^	1.19 ± 0.31	1.31 ± 0.27 ^†^
LVPWd (mm)	0.81 ± 0.06	0.84 ± 0.05	0.86 ± 0.04	0.93 ± 0.16	0.77 ± 0.07	0.86 ± 0.07
LVPWs (mm)	1.44 ± 0.07	1.09 ± 0.06 *	0.96 ± 0.07 *	0.80 ± 0.10 *^†^	0.89 ± 0.10 *	0.97 ± 0.15 *
FS (%)	62.2 ± 3.5	24.5 ± 4.0 *	51.1 ± 3.7 *	32.0 ± 3.85 *^†&^	56.4 ± 6.4	51.6 ± 6.6 ^†^
EF (%)	85.2 ± 2.4	42.5 ± 6.3 *	75.7 ± 3.7 *	53.3 ± 5.01 *^†&^	80.0 ± 4.9	74.8 ± 7.4 ^†^
Hypokinesia (%)	0 (n = 0/6)	60 (n = 3/5) *	20 (n = 1/5)	20 (n = 1/5)	0 (n = 0/5)	50 (n = 3/6) *
Dyskinesia (%)	16.7 (n = 1/6)	20 (n = 1/5)	0 (n = 0/5)	20 (n = 1/5)	20 (n = 1/5)	50 (n = 3/6)
Aneurysm (%)	0 (n = 0/6)	20 (n = 1/5)	0 (n = 0/5)	0 (n = 0/5) ^&^	0 (n = 0/5)	66.7 (n = 4/6) *
**ECG**						
HR (bpm)	440 ± 18	494 ± 17 *	461 ± 7	524 ± 26 *	472 ± 26	501 ± 9.3 *
RR-I (ms)	137 ± 5.7	121 ± 4.6 *	130 ± 2.0	115 ± 5.9 *	128 ± 7.3	120 ± 2.4 *
PR-I (ms)	36.7 ± 1.2	36.3 ± 0.76	39.1 ± 0.74 *	40.9 ± 2.2 *	34.9 ± 1.3	36.5 ± 2.2
Pd (ms)	11.3 ± 0.56	9.70 ± 1.8	12.0 ± 0.56	10.5 ± 0.23	11.0 ± 1.2	12.1 ± 1.1
QRSd (ms)	12.7 ± 0.28	18.9 ± 5.2	11.8 ± 0.28 *	12.1 ± 2.9	11.9 ± 0.92	17.8 ± 3.1 *
Q-Amp (mV)	−0.08 ± 0.02	−0.12 ± 0.04	−0.02 ± 0.00 *	−0.09 ± 0.06 ^&^	−0.08 ± 0.01	−0.22 ± 0.05 *^†^
S-Amp (mV)	−0.21 ± 0.04	−0.01 ± 0.02 *	−0.26 ± 0.06	−0.01 ± 0.04 *	−0.17 ± 0.11	−0.03 ± 0.02 *
PVCs (%)	0.0 ± 0.0	0.20 ± 0.23	0.05 ± 0.06	0.27 ± 0.08 *^&^	2.3 ± 2.2	9.4 ± 4.1 *^†^
**BPP**						
SBP (mmHg)	108 ± 2.8	124 ± 5.9 *	101 ± 3.8	106 ± 7.8 ^&^	95.5 ± 4.0 *	85.0 ± 9.4 *^†^
DBP (mmHg)	56.8 ± 5.3	65.2 ± 4.6	58.6 ± 8.6	50.4 ± 4.7 ^†^	51.5 ± 3.2	44.8 ± 4.0 ^†^
MBP (mmHg)	74.2 ± 4.4	84.8 ± 4.8	74.4 ± 6.9	69.0 ± 5.2 ^†&^	66.0 ± 2.4 *	58.2 ± 5.8 ^†^
**Morphometric**						
RWT (mm)	0.67 ± 0.06	0.58 ± 0.06	0.69 ± 0.07	0.64 ± 0.10	0.60 ± 0.09	0.67 ± 0.07
LVM (mg)	65.0 ± 6.9	82.9 ± 8.2	61.2 ± 7.0	84.3 ± 15.8	58.8 ± 13.4	59.8 ± 6.0 ^†^
LVW/BW (mg/g)	2.73 ± 0.27	3.39 ± 0.28	2.44 ± 0.14	3.65 ± 0.85	2.53 ± 0.25	2.59 ± 0.24 ^†^
LW/BW (mg/g)	45.3 ± 1.7	44.6 ± 1.1	49.7 ± 1.3 *	50.8 ± 1.5 *^†^	42.3 ± 2.0	36.8 ± 7.1
HW/BW (mg/g)	4.68 ± 0.06	4.94 ± 0.17	4.35 ± 0.10 *	5.10 ± 0.25 *^&^	4.22 ± 0.20 *	4.52 ± 0.13 ^†^

## Data Availability

The data presented in this study are available on request from the corresponding author. The data are not publicly available due to privacy.

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
