# Peer review of "Efficacy and Safety of Angiotensin Receptor Blockers in a Pre-Clinical Model of Arrhythmogenic Cardiomyopathy"

_ijms, 2022, doi:10.3390/ijms232213909_

Round 1
Reviewer 1 Report
Comments to the author:
The manuscript entitled “Efficacy and Safety of Angiotensin Receptor Blockers in a Pre-Clinical Model of Arrhythmogenic Cardiomyopathy Efficacy and Safety of Angiotensin Receptor Blockers in a Pre-Clinical Model of Arrhythmogenic Cardiomyopathy”. This study analyzed that Telmisartan-treated Dsg2mut/mut mice showed increased cardiac wall motion abnormalities, augmented %PVCs, electrocardiographic repolarization/depolarization abnormalities, larger fibrotic lesions, and increased expression of PPARy-regulated gene transcripts compared to their Dsg2mut/mut counterparts. Alternatively, Valsartan-treated Dsg2mut/mut mice harbored fewer myocardial scars, reduced %PVC, and increased Wnt-mediated transcripts. Considering our findings, caution should be taken by physicians in prescribing medications that may increase PPARy signaling in patients with ACM. It need some corrections before acceptance.
Comments
1- Please measure protein expression I figure no 2H which will give right information about PPAR signalling in Dsg2mut/mut mice heart.
2- Please add the information of cre, which cre was used to make Dsg2mut/mut.
Reviewer 2 Report
Summary: The main aim of the authors in this study is to determine the safety of angiotensin receptor blockers in comparison with the agents that may trigger PPARy signaling such as Rosiglitazone. The authors conducted in vivo experiments on Desmoglein-2 mutant mice (Dsg2mut/mut) mice to test the two angiotensin receptor blockers Valsartan and Telmisartan and Rosiglitazone peroxisome proliferator-activated receptor gamma (PPARγ). Cardiac function, premature ventricular contractions, fibrofatty scars, PPARα/γ protein levels and PPAR-mediated mRNA transcripts were assessed in these animals. The authors stated from their findings that, none of the mice treated with the PPARy agonists reached the end point of the study whereas the animals treated with ARBs such as telmisartan has shown a better contractile function compared to the vehicle. Valsartan ,another ARB although did not delay they the progression of cardiac dysfunction but it normalized the QRS duration, reduced ventricular ectopy and prevented fibrotic progression. The authors concluded that the physicians must be vigilant which prescribing the medication that may trigger or increase PPARy signaling in the patients with arrhythmogenic cardiomyopathy.
Comments:
1. There are currently other angiotensin receptor blockers that could be prescribed for ACM by the physicians. Could the authors explain the reason for choosing Valsartan and Telmisartan for this study?
2. In an article commentary by Strauss et al., (Circulation,2006) the authors argue that ARBs may increase the risk of myocardial infraction. The reviewer is curious to know if the authors have performed any histological staining to evaluate the heart tissue for myocardial infraction.
3. The reviewer believes that the conclusion could be further strengthened by adding additional number of mice to the treatment groups.
